# Green returns to education: Does education affect pro-environmental attitudes and behaviors in China?

**Qi Wang[1], Geng Niu[2], Xu Gan[3]\*, Qiaoling Cai[2]**

**1** Business School, Sichuan University, Chengdu, Sichuan, P.R. China, **2** Research Institute of Economics and Management, Southwestern University of Finance and Economics, Chengdu, Sichuan, P.R. China, **3** School of Management, Huazhong University of Science and Technology, Wuhan, Hubei, P.R. China

\* xgan@hust.edu.cn

**Data Availability Statement:** Data cannot be shared publicly because it is owned by a third party and authors do not have permission to share the data. Data are available from the Chinese General Social Survey (CGSS) database (contact via http://cgss.ruc.edu.cn/English/Home.htm) for

## Abstract

Using microdata from the Chinese General Social Survey (CGSS 2010), this paper investigates whether there are green returns to education in China, where educational attainment promotes pro-environmental attitudes and behaviors. We establish causality by exploiting the exogenous variation induced by the implementation of the Compulsory Schooling Law (CSL) in China. We find evidence that educational attainment is associated with higher levels of pro-environmental attitudes and behaviors, and these estimates are robust to various robustness checks. Further analysis reveals that the acquisition of environmental knowledge is the channel that drives the effect of education on pro-environmental attitudes and behaviors. Finally, the effects of education are heterogeneous across individuals.

## 1. Introduction

China faces severe environmental problems such as air pollution, depletion of resources, extreme weather events and food contaminations in recent years [1–3]. Environmental problems not only significantly increase the morbidity and mortality rates of various diseases but also generate a heavy medical burden and large disutility [4]. The health loss erodes average life expectancy and human-capital accumulation, thereby jeopardizing economic development [5]. These environmental problems are, at least partly, rooted in human behavior [6,7], and can thus be mitigated by changing the relevant behaviors so as to improve environmental quality. Therefore, understanding the determinants of pro-environmental attitudes and behaviors is of high academic and policy relevance in China. Nevertheless, the research in this direction is still in its infancy. Most of the existing studies have focused on the harmful effects of pollutions produced by firm-level manufacturing activity in China [8]. In fact, an important part of the environmental contamination comes from the daily practices of the households, which has not received sufficient scholarly attention [9].

This paper undertakes an in-depth examination of the role that education plays in affecting individual pro-environmental attitudes and behaviors in China. The association between education and many positive economic, social and health outcomes is well documented in the

researchers who meet the criteria for access to confidential data. Interested researchers can replicate our study findings in their entirety by directly obtaining the data from the third party listed above and following the protocol. In addition, we confirm that the authors of this paper did not have any special access privileges that others would not have.

**Funding:** This work was supported by the 111 Project of China (No. B16040) and the National Natural Science Foundation of China (No. 71904160). The funders had no role in study design, data collection and analysis, decision to publish, or preparation of the manuscript.

**Competing interests:** The authors have declared that no competing interests exist.

economic literature [10–14]. In principal, education may affect household environmental behaviors in several ways. First, formal schooling is the primary way individuals acquire knowledge and skills [15]. Second, education helps people to obtain access to more sources and types of information [16], which further facilitates knowledge acquisition. Knowledge and information about environmental issues might increase individuals' sense of environmental responsibility. In addition, education might increase the marginal utility of pro-environmental behaviors by enabling people to pursue a higher level of demand [17]. Alternately, education might decrease people's environmental awareness if the education system directs people to be obsessed with competition and thus makes people focus on self-enhancement values [18].

In this paper, we rely on data from the Chinese General Social Survey (CGSS) conducted in wave 2010 for empirical analysis. A major advantage of this dataset lies in the provision of questions concerning a wide range of pro-environmental attitudes and behaviors. We find a strong positive association between education and pro-environmental attitudes as well as behaviors. To alleviate the potential endogenous concern, we instrument education with personal exposure to the implementation of the Compulsory Schooling Law (henceforth, CSL) in China. The two-stage least squares (2SLS) regression results confirm that education imposes a positive influence on both pro-environmental attitudes and behaviors. In particular, an additional year of schooling results in an increase of 0.012 and 0.011 in the pro-environmental attitudes and behaviors, which are approximately 2.1% and 2% of the corresponding sample means. To test the validity of our analysis, we conduct several robustness checks. It turns out that our results are robust to placebo tests, alternative measures of pro-environmental attitudes and behaviors, and sample restrictions. We also find that the effects of education on pro-environmental attitudes and behaviors are more pronounced for males, rural residents, and low-income individuals.

In addition, we examine a potential mechanism underlying the effect of schooling education on pro-environmental attitudes and behaviors, namely environmental knowledge. The willingness to alleviate environmental problem is a partial function of risk perception, which is reflected in environmental knowledge [5]. The acquisition of environmental knowledge makes individuals more informed about the situation of local and global environment, negative health effects of environmental problems, and public environmental policies. Indeed, education facilitates a better understanding of complex environmental messages such as climate change [15,19,20]. Hence, the environmental knowledge may play a major role in forming people's pro-environmental attitudes and behaviors. Thanks to the richness of the CGSS 2010 dataset, we are able to measure the level of environmental literacy among Chinese households and test whether environmental knowledge is an underlying mechanism through which formal education influences pro-environmental attitudes and behaviors. In particular, the CGSS 2010 asks ten true or false questions concerning fundamental concepts of environment to assess the respondents' environmental knowledge. We observe a lack of environmental literacy among Chinese people. The proportion of correct answers to most of questions is below 62%, while the proportion of the corresponding incorrect responses or "do not know" answers is quite sizable. We also discover that people with a higher level of formal education are more likely to get higher accuracy rate in answering environmental questions. Furthermore, a mediation analysis is conducted to investigate whether environmental knowledge serves as a mediator. We find that the effect of education on pro-environmental attitudes and behaviors attenuates substantially after controlling for environmental knowledge.

The contributions of this paper are three-fold. First, we add to the small yet burgeoning literature on household environment-related behaviors [5,7,21] by identifying the important role that formal education plays in shaping pro-environmental attitudes and behaviors. Furthermore, we examine the pro-environmental attitudes and behaviors of households in the

context of the severe environmental problems in China. While there are a large number of papers studying the pollutions associated with firm-level manufacturing activities in China [22,23], the literature on Chinese households' pro-environmental behaviors is rather underdeveloped. In particular, with the increase in social responsibility for environmental well-being and the sustainable development of economy, Chinese households are encouraged to take pro-environmental behaviors. For example, following the practice in Shanghai, many local governments in China are considering introducing new garbage-sorting regulations. Thus, it is important to understand the determinants of pro-environmental attitudes and behaviors in China, which is the theme of our study.

Second, despite the ample evidence on the association between education and pro-environmental behaviors including consumption, conservation and lifestyle [1,24,25], most of the existing studies fail to deal with the potential endogeneity of education. The only exceptions are Meyer [26] and Chankrajang and Muttarak [15], both of which employ changes in compulsory education laws to establish a causal relation between education and pro-environmental behaviors in Europe and in Thailand, respectively. Building on their works, we are among the first to provide causal estimates of the effects of general education on pro-environmental attitudes and behaviors in China. Studying the Chinese context is interesting and important. As the cost of fast industrial growth and rapid urbanization, China has suffered a series of environmental problems [27]. As the largest developing country in the world, the environmental problems of China have strong global impacts. Environmental attitudes and behaviors of Chinese residents play an important role in affecting the country's future environmental scenarios. In addition, we identify that the acquisition of environmental knowledge partly mediates the causal effect of education on pro-environmental attitudes and behaviors. The existing studies have found that education has speeded up the acquisition and the diffusion of knowledge [28]. Our study therefore contributes to the literature by shedding direct lights on the channel behind the nexus between education and pro-environmental behaviors.

Third, our study contributes to the literature that explores the social returns to education. Previous works have investigated the impact of schooling education on economic, social and health outcomes such as labor market returns [29], risk-takings in financial market [14], climate adaptation [28], adult civic engagements and attitudes [10], cognitive abilities at old age [12], health capital [11] and mortality [13]. In particular, utilizing the firm level data, Amore et al. [21] show that CEO education forms managerial styles, giving rise to greater sustainability in corporate actions. By providing a detailed assessment at individual-level of how formal education influences pro-environmental attitudes and behaviors in China, we lend further support to those who stress the importance of education's positive outcomes. Our results indicate that the improvement of educational attainment promotes pro-environmental attitudes and behaviors, which in turn are likely to have a positive influence on economic sustainability.

Our study also has important policy implications. As in most industrialized and industrializing nations, Chinese government is seeking to set the country on a path to more sustainable development, with a focus on the effectiveness administrative and regulatory instruments [30]. This paper offers an alternative solution by stressing the efficiency of increasing access to general education as an essential way of motivating pro-environmental attitudes and behaviors among a broad population.

The rest of the paper proceeds as follows: Section 2 introduces the related literature and the background of the CSL in China. Section 3 describes the data source and estimation strategy. Section 4 presents the main results and the robustness of our results. Section 5 provides the possible mechanism analysis. Section 6 summarizes the results and provides some concluding thoughts.

## 2. Literature review and institutional background

It is well documented that education is associated with a wide range of positive outcomes [31]. While early studies focus on financial outcomes such as income and wealth, the non-pecuniary returns to education, in various domains such as job satisfaction, health, marriage, and parenting, are receiving increasing scholarly attention [31]. Moreover, education not only brings private returns but might also generate externalities by affecting social outcomes such as economic growth, innovation, regional crime, and tax revenue [32–34].

As environmental problems are closely related to human behavior [6,7], individuals' pro-environmental attitudes and behaviors are expected to bring positive externalities to the society. Education might promote pro-environmental attitudes and behaviors in several ways. First, formal schooling is the primary way individuals acquire knowledge and skills [15]. As environmental concepts and topics can be complicated, knowledges and skills accumulated through education make individuals familiar with scientific terms in environment issues [15,19]. Second, and relatedly, people with a higher level of formal education have access to more sources and types of information [16]. An increased exposure to information further increases peoples' environmental knowledge and environmental awareness. Third, based on the hierarchy of needs theory [17,35], education might help individuals meet their basic needs, and thereby enabling them to adopt a pro-environmental lifestyle to pursue a higher level of demand [9,36]. In another word, education might increase the marginal utility of pro-environmental behaviors relative to the marginal utility of money.

While some prior studies have explored the relation between education and pro-environmental behaviors [1,15,24–26], causal evidence in the context of China is still lacking. A causal analysis can provide important public policy implications. China also offers an interesting context to study this topic. China is facing many severe environmental problems [1–3]. As the second largest economy in the world, China' environmental problems can have global impacts. For example, China has been ranked first of world top economies in terms of carbon dioxide emissions [37].

Since the 1980s, China began a process of education reform that aimed to gradually align the educational system with the newly emerging marketization of the economy. The Compulsory Schooling Law (CSL) of the People's Republic of China was passed on April 12, 1986 and officially went into effect on July 1, 1986. This was the first formal law to specify education policies for the entire country [38–40]. The CSL stipulated that a nine-year compulsory education should be mandatory and free for school-age children in China and across all ethnicity groups. All children at age of six (or sometimes age seven) were required to enroll in schools. A nine-year compulsory education was divided into two stages: six-year primary education and three-year junior middle school education. After primary school, graduates can be directly admitted into junior middle schools without general selection tests. Additionally, it was illegal to employ school-age children or adolescents for any organization or individual. Since local governments are allowed to autonomously determine the steps to implement the CSL in accordance with the economic development of the region [41], the timing of the actual implementation of the CSL varies significantly across provinces. S1 Table in S1 Appendix shows the time when the CSL was introduced at the province level. Some coastal provinces or municipalities were among the first to enforce the reform (e.g., Shanghai, Beijing and Zhejiang), while other provinces did not implement the new system until the early or mid-1990s (e.g., Gansu, Guangxi, Hainan, Hunan and Tibet). The overall year gap is six years between the earliest provinces and latest provinces in the sample.

The CSL in China notably raised the average education level of a broad population. By 2000, China basically realized the goal of universal compulsory education, covering 85% of its

population. As illustrated in Fig 1, the junior secondary school enrollment rate dramatically increased from 68.4% in 1985 to 98.4% in 2005. Meanwhile, the illiteracy rate fell from 23.5% in the 1982 to 6.72% in the 2000 according to national census. In the meantime, the implementation of the CSL is a great exogenous factor with respect to personal educational attainment in China, because it is independent of individual preferences and economic conditions and cannot be adjusted in advance. Hence, we utilize this shock to construct an instrumental variable for education, which will be explained in detail in the identification strategy section.

## 3. Data, variables, and models

### 3.1. Data

Our main source of data is the Chinese General Social Survey (CGSS), which is a comprehensive database that aims to systematically monitor the changing relationship between social structure and quality of life in both urban and rural China. Initiated in 2003, the CGSS is one of the earliest national representative cross-sectional survey projects in China. Since 2010, the survey has been conducted annually by the National Survey Research Center (NSRC) at Renmin University of China. The respondents are required to be age 18 and above.

We use the 2010 round of CGSS data to explore whether there are green returns to education in China, because it is the only wave that contains information about respondents' pro-environmental attitudes and behaviors as well as environmental knowledge, and is in the meantime publicly available. In 2010, the survey covers 11,783 respondents, residing in 134 cities across 31 of China's provinces. To shorten questionnaire length and improve survey compliance, some modules in CGSS 2010 are randomly assigned to a subset of respondents. For example, for questions on environmental attitudes and behaviors, only those born in February, September, November, or December are asked. Thus, our study can only use approximately one third of the sample. We dropped the observations with missing values for variables that are used in the analysis and end up with 3,661 observations. We conduct a balancing test based on a range of observational variables between the regression sample and the dropped sample to check whether there exists a severe sample selection issue. The results, shown in S2 Table in S2 Appendix, suggest that the two samples are similar in observational characteristics, which gives us further confidence in the representativeness of our regression sample.

### 3.2 Variables

**3.2.1 Pro-environmental attitudes and behaviors.** The CGSS 2010 asks respondents three questions regarding the willingness of their pro-environmental attitudes (Question 12 in L part of the survey) and six questions about the frequency of their pro-environmental behaviors (Question 20 in L part of the survey). For three attitude questions, respondents are asked to rate their willingness to pay higher prices, higher taxes and to lower their standard of living for environmental protection based on five ordinal scales: (i) "Extremely unwilling to"; (ii) "Reluctant"; (iii) "Do not care"; (iv) "Willing to" and (v) "Extremely willing to". As for the six pro-environmental behaviors, taking waste recycling for example, the respondents are asked whether they have taken any actions to separate wastes for recycling, and they need to indicate how often they carry out such actions given three ordinal options: (i) "Not at all", (ii) "Occasionally" and (iii) "Frequently". The exact wording of other environmental behaviors is provided in Table 1.

Table 1 summarizes the pro-environmental attitudes and behaviors. Specifically, Panel A reports the proportion of respondents providing "Extremely unwilling", "Reluctant to", "Do not care", "Willing to" and "Extremely willing to" answers to each of the three pro-environmental attitudes questions, while Panel B shows the distribution of respondents providing

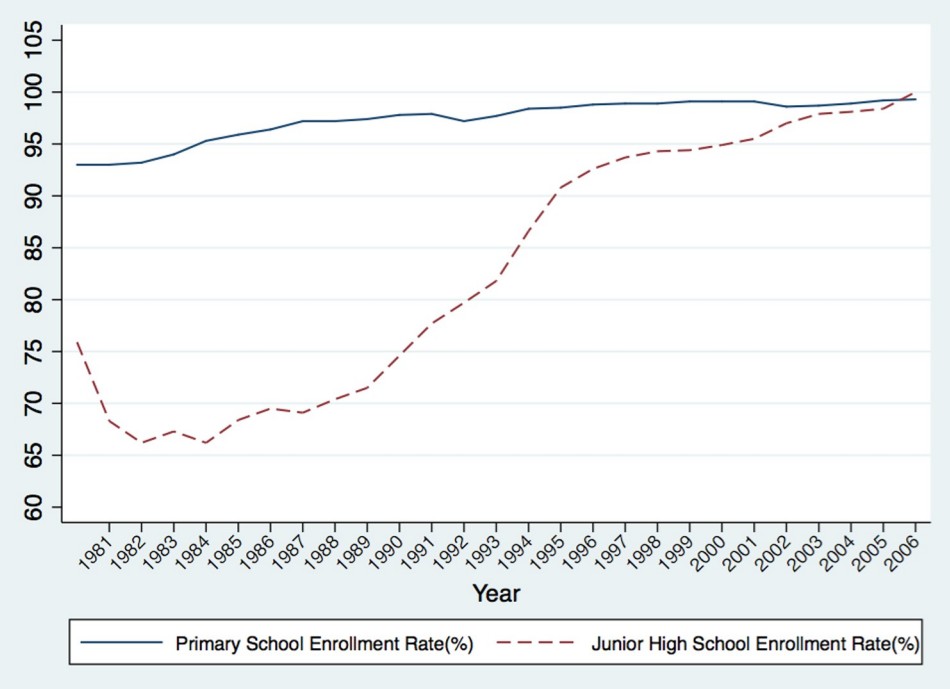

**Fig 1. School enrollment rate in China 1981–2006.** Note: This figure illustrates the school enrollment rate in China over the period between 1981 and 2006. The solid line represents the primary school enrollment rate and the dashed line represents the junior high school enrollment rate. Data source is National Bureau of Statistics of China.

"Not at all", "Occasionally", and "Frequently" answers to each of the six pro-environmental behaviors questions. Overall, there are significant variations in respondents' pro-environmental attitudes. To protect the environment, respondents are more willing to pay higher prices than paying higher taxes and lowering living standard. Panel B of Table 1 shows that the majority of Chinese respondents carry out pro-environmental behaviors on a regular basis.

Following the previous literature [26], we aggregate the three attitudes questions into one variable. For each of the three attitude questions, respondents are asked to provide answers on a five-point scale from "Extremely unwilling to" (i) to "Extremely willing to" (v), with a higher value indicating a stronger pro-environmental attitude. By summing answers to the three questions together, we obtain a variable with a maximum score of 15. We divide this variable by 15 to construct the variable *Pro-environmental attitudes*, which has a maximum score of one. Similarly, we aggregate the six behavior questions into one variable. For each of the six behavior questions, respondents provide answers on a three-point scale, with a higher value indicating a higher intensity of pro-environmental behaviors. We construct the variable *Pro-environmental behaviors* by summing answers to the six behavior questions and divide the result by 18, which makes the variable has a maximum value of one. All else being equal, individuals who have higher values of pro-environmental attitudes and behaviors are likely to think and behave in a more environmentally friendly manner.

**3.2.2 Other variables.** The main explanatory variable for our analysis is *Education*, which is defined as the years of schooling that the respondent receives as in the literature [38,42]. Since the CGSS does not directly ask for the years of schooling, we quantify *Education* by assigning zero to being without schooling, six years to a primary school education, nine years to a junior high school education, twelve years to a senior high school education, thirteen years to a vocational school education, fifteen years to a junior college education, sixteen years

**Table 1. Summary of pro-environmental attitudes and behaviors.**

| | | | | | % of respondents |
|---|---|---|---|---|---|
| Panel A: Pro-environmental attitudes | | | | | |
| | Extremely unwilling to | Reluctant to | Do not care | Willing to | Extremely willing to |
| To protect the environment, are you willing to pay higher prices? | 7.8 | 25.4 | 20.2 | 37.3 | 9.4 |
| To protect the environment, are you willing to pay higher taxes? | 9.8 | 31.2 | 21.1 | 31.7 | 6.2 |
| To protect the environment, are you willing to lower your standard of living? | 12.6 | 33.7 | 19.6 | 28.6 | 5.4 |
| Panel B: Pro-environmental behaviors | | | | | |
| | | | Not at all | Occasionally | Frequently |
| How often do you separate wastes for recycling? | | | 23.8 | 32.6 | 43.5 |
| How often do you buy locally produced products or groceries without chemical fertilizers or pesticides? | | | 32.0 | 37.7 | 30.3 |
| How often do you buy environmentally friendly products marked with an environmental label? | | | 34.1 | 41.6 | 24.3 |
| How often do you reduce car usage for environmental protection? | | | 28.8 | 44.4 | 26.9 |
| How often do you cut down energy consumption for environmental protection? | | | 27.1 | 40.3 | 32.7 |
| How often do you cut down water consumption for environmental protection? | | | 17.0 | 34.0 | 49.0 |

Note: Panel A reports the proportion of respondents providing "Extremely unwilling", "Reluctant to", "Do not care", "Willing to" and "Extremely willing to" answers to each of the three pro-environmental attitudes questions. Panel B reports the proportion of respondents providing "Not at all", "Occasionally", and "Frequently" answers to each of the six pro-environmental behaviors questions. Percentages may not sum up to 100 due to rounding.

to a college level, eighteen years to a graduate school education and beyond. Moreover, the CGSS data set collects detailed demographic information, which allows us to control for a range of confounding factors. Specifically, we control for *Male*, *Married*, *Income*, *Employed*, *Han nationality*, *Rural*, cohort fixed effect, and city fixed effect. Table 2 provides the definitions of variables of interest.

Table 2 reports the descriptive statistics of the variables outlined above. The mean educational attainment is 8.933 years, which suggests that on average the final degree that the respondents in our sample receive is merely junior high school. The majority of respondents are married, employed and belong to Han ethnic nationality. 47.3% of respondents are men and 48.8% live in rural areas.

To have a preliminary view on how education relates to pro-environmental attitudes and behaviors, we plot the average educational attainment and the average pro-environmental

**Table 2. Descriptive statistics.**

| Variables | Definition | Obs. | Mean | Std. Dev. | Median | Min | Max |
|---|---|---|---|---|---|---|---|
| Pro-environmental attitudes | The intensity of an individual's pro-environmental attitudes | 3,458 | 0.569 | 0.210 | 0.6 | 0.066 | 1 |
| Pro-environmental behaviors | The intensity of an individual's pro-environmental behaviors | 3,661 | 0.539 | 0.199 | 0.555 | 0.055 | 1 |
| Education | Years of schooling | 3,661 | 8.933 | 4.607 | 9 | 0 | 18 |
| Age | Age in number of years | 3,661 | 47.31 | 15.73 | 47 | 18 | 85 |
| Male | = 1 if male, = 0 otherwise | 3,661 | 0.473 | 0.499 | 0 | 0 | 1 |
| Married | = 1 if married, = 0 otherwise | 3,661 | 0.807 | 0.395 | 1 | 0 | 1 |
| Income | Ln (Personal income+1) | 3,661 | 9.298 | 4.093 | 9.473 | 0 | 16.118 |
| Han nationality | = 1 if Han nationality, = 0 otherwise | 3,661 | 0.918 | 0.275 | 1 | 0 | 1 |
| Employed | = 1 if employed, = 0 otherwise | 3,661 | 0.629 | 0.483 | 1 | 0 | 1 |
| Rural | = 1 if rural, = 0 otherwise | 3,661 | 0.488 | 0.500 | 0 | 0 | 1 |
| CSL | Exposure to implementation of the CSL | 3,661 | 0.222 | 0.386 | 0 | 0 | 1 |
| Environmental knowledge | Number of correct responses to ten environmental knowledge questions | 3,601 | 5.154 | 2.762 | 5 | 0 | 10 |

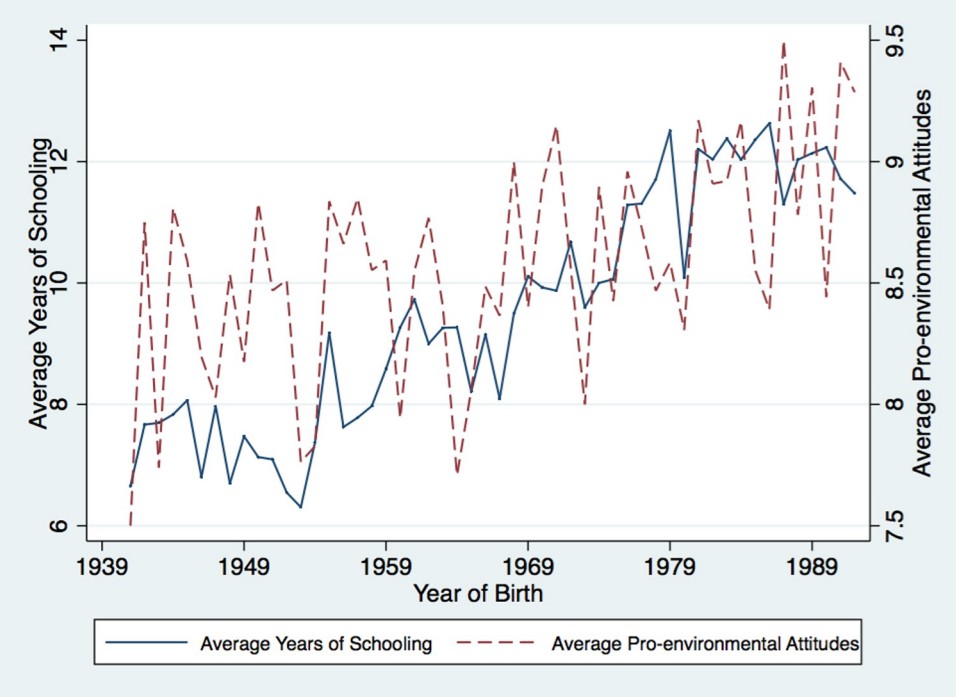

**Fig 2. Average years of schooling and pro-environmental attitudes across year of birth.** Note: This figure illustrates the relationships between education and pro-environmental attitudes and behaviors across year of birth. The solid line represents the average years of schooling and the dashed lines represent the average pro-environmental attitudes (2a) and behaviors (2b) respectively. Data source is CGSS 2010.

attitudes and behaviors against the birth-year level in Figs 2 and 3. Obviously, the educational level displays an upward trend; it increases from 6.66 years in 1930 to 12.23 years in 1990. More importantly, the average pro-environmental attitudes and behaviors move very closely with the level of education, suggesting strong positive correlations between education and pro-environmental attitudes and behaviors, respectively.

To address the possibility that education is correlated with other individual characteristics and to gauge the impact of education on pro-environmental attitudes and behaviors more precisely, we proceed with a regression framework, which is specified below.

### 3.3 Model specifications

We estimate the relationship between education and pro-environmental attitudes and behaviors using the following multivariate model:

$$Y_{ij} = \alpha + \beta' Education_{ij} + \delta' X_{ij} + \gamma C_h + \rho' City_k + \varepsilon_{ij}, \tag{1}$$

where $Y_{ij}$ is the pro-environmental attitudes and behaviors for individual $i$ in province $j$, $Education_{ij}$ is the years of schooling, $X_{ij}$ is a vector of control variables, $\alpha$ is the intercept, $C_h$ is a vector of cohort dummies, and $\varepsilon_{ij}$ is the error term. Given the distinct regional segregation in China, we add city dummies $City_k$ in all regressions to capture city fixed effects. We are mainly interested in the coefficient of $Education_{ij}$, β, which directly captures the relationship between educational attainment and pro-environmental attitudes and behaviors. We use OLS for estimation and report standard errors clustered at the cohort-city level to adjust for the potential correlation of observations within the same cluster.

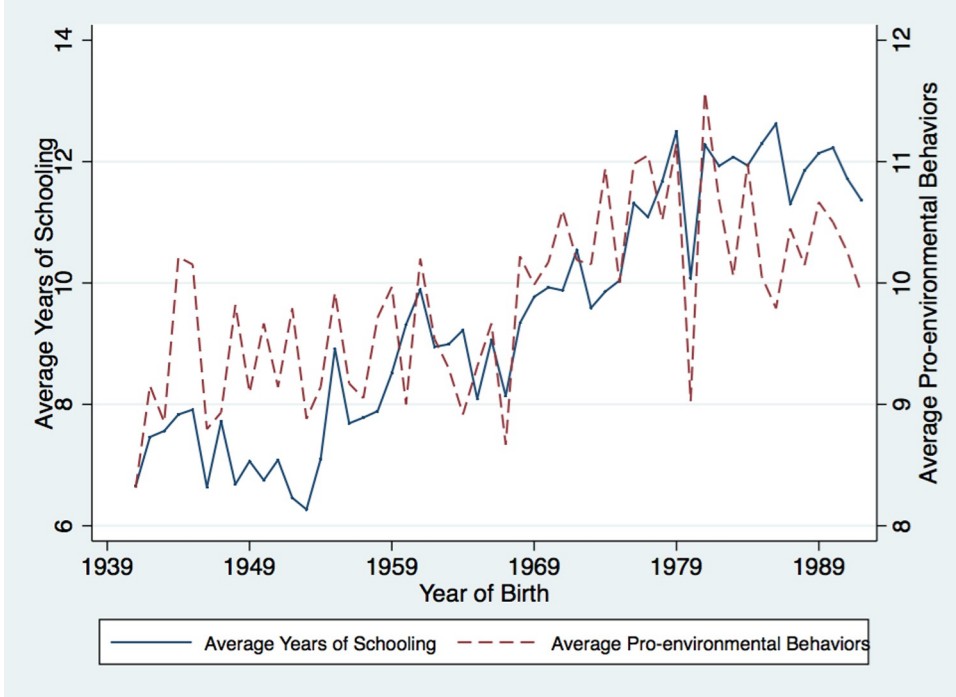

**Fig 3. Average years of schooling and pro-environmental behaviors across year of birth.** Note: This figure illustrates the relationships between education and pro-environmental behaviors across year of birth. The solid line represents the average years of schooling and the dashed lines represent the average pro-environmental behaviors. Data source is CGSS 2010.

Based on the above OLS regression, however, we cannot give a causal interpretation for the relationship between educational attainment and pro-environmental attitudes and behaviors for the omitted variable bias due to the unobservable confounding factors. For example, some individual characteristics that can affect pro-environmental attitudes and behaviors are hard to measure, e.g., sense of responsibility. Those with higher sense of responsibility are more likely to achieve higher educational attainment and in the meantime care more about environmental issues. If this is the case, the OLS estimates would be biased.

To address the endogeneity problem, we adopt an instrumental variable approach to estimate the causal effects of education on pro-environmental attitudes and behaviors. Valid instruments need to be correlated with the endogenous variable and not to be linked to the dependent variable through channels other than the independent variable. We exploit the enactment of the CSL in 1986 as an exogenous source of changes in individual schooling levels. On the one hand, as discussed above, the implementation of the CSL in China made a great achievement in increasing the average education level of the population as a whole. This implies that individuals that are more affected by the CSL are likely to end up with higher educational attainment. On the other hand, the enactment and implementation of the CSL are exogenously determined by governments and are unlikely to be correlated with personal characteristics and other potentially confounding factors that influence pro-environmental attitudes and behaviors. Moreover, the CSL would not directly affect pro-environmental attitudes and behaviors, because the development of pro-environmental concerns hardly started at that time.

Following Liang and Dong [42], we construct the instrumental variable, *CSL*, by measuring individual exposure to the implementation of compulsory schooling law. We define it as a

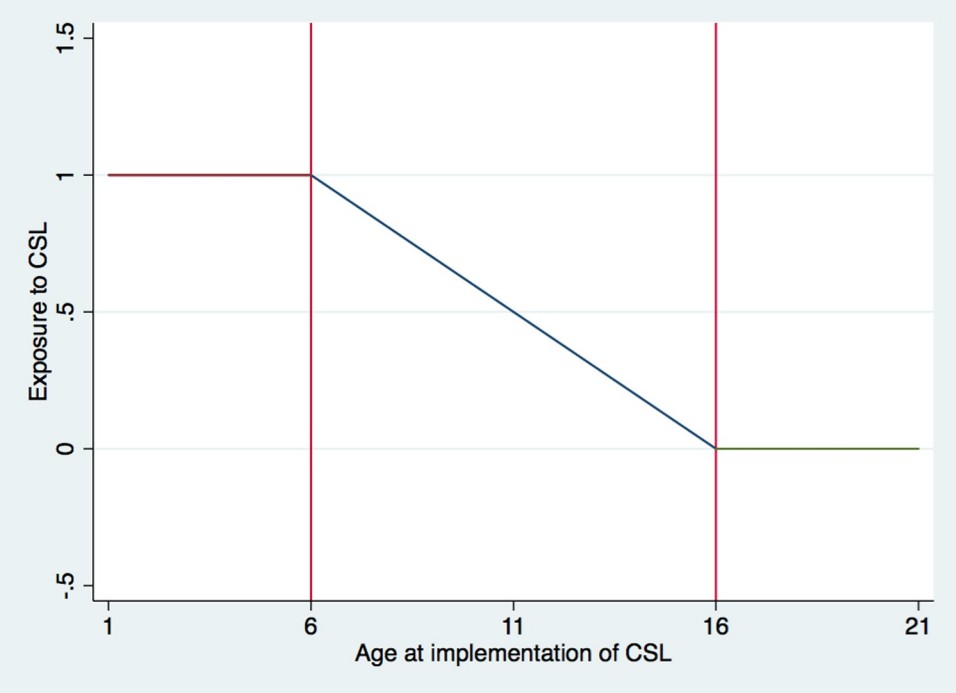

**Fig 4. Construction of instrumental variable.** Note: This figure illustrates the construction of instrumental variable. The X-axis is the individual age at the implementation of the CSL and the Y-axis is the individual exposure to CSL, which equals to one if the individual is fully affected by CSL (i.e. aged 6 or below) and equals to zero if the individual is unaffected by CSL (i.e. age 16 or above). A linear function is assumed for the ages in between.

continuous variable that falls between zero and one. Specifically, *CSL* equals one if the individual is fully affected by the compulsory schooling law (i.e. aged 6 or below) and equals zero if he/she is completely unaffected by the law (i.e. aged 16 or above). For those whose ages were between 6 and 16 when the law was enforced, we define *CSL* as (16 –year of implementation + year of birth)/9, a decreasing linear function of age. Fig 4 illustrates the construction of the instrumental variable. Note that in constructing the instrumental variable, we take into account the fact that the year of implementation varies across provinces as shown in S1 Table in S1 Appendix. In other words, for individuals of the same age, their exposures to the CSL may differ due to the differences in the timing of the CSL in the provinces they lived.

Using the instrument *CSL*, we re-estimate the effect of education on pro-environmental attitudes and behaviors by the two-stage least squares (2SLS) approach:

$$Education_{ij} = \alpha_0 + \alpha_1 CSL_{ij} + \alpha_2 X_{ij} + \alpha_3 C_h + \alpha_4 City_k + \eta_{ij}, \tag{2}$$

$$Y_{ij} = \beta_0 + \beta_1 \widehat{Education}_{ij} + \beta_2 X_{ij} + \beta_3 C_h + \beta_4 City_k + e_{ij}, \tag{3}$$

where Eqs (2) and (3) are the first stage and second stage regressions, respectively. $\widehat{Education}_{ij}$ represents the fitted value of $Education_{ij}$ from the first stage regression, and the other variables are the same as in Eq (1). Again, we cluster standard errors at the cohort-city level.

**Table 3. The effects of education on pro-environmental attitudes and behaviors, baseline results.**

| | Pro-environmental attitudes | | Pro-environmental behaviors | |
|---|---|---|---|---|
| | **(1)** | **(2)** | **(3)** | **(4)** |
| Education | 0.009*** | 0.008*** | 0.011*** | 0.008*** |
| | (0.001) | (0.001) | (0.001) | (0.001) |
| Male | | 0.005 | | -0.001 |
| | | (0.008) | | (0.005) |
| Married | | 0.009 | | 0.011 |
| | | (0.009) | | (0.008) |
| Income | | 0.001 | | 0.001 |
| | | (0.001) | | (0.001) |
| Employed | | -0.008 | | -0.010 |
| | | (0.007) | | (0.007) |
| Han nationality | | -0.058 | | -0.017 |
| | | (0.022) | | (0.016) |
| Rural | | -0.012 | | -0.063*** |
| | | (0.011) | | (0.010) |
| Cohort FE | Yes | Yes | Yes | Yes |
| City FE | Yes | Yes | Yes | Yes |
| Observations | 3,458 | 3,456 | 3,664 | 3,661 |
| R-Square | 0.177 | 0.194 | 0.301 | 0.316 |

Note: The table reports the estimates from OLS regressions of education on pro-environmental attitudes and behaviors. Columns (1)-(2) report the results for pro-environmental attitudes, while columns (3)-(4) report the results for pro-environmental behaviors. All regressions include a constant term, cohort fixed effects, and city dummies, but their coefficients are suppressed for brevity. Robust standard errors are clustered at the cohort-city level and are reported in parentheses. ***, ** and * indicate 1%, 5% and 10% significance levels, respectively.

## 4. Regression results

### 4.1. Benchmark results

We first examine the relationship between educational attainment and pro-environmental attitudes and behaviors by estimating Eq (1) using OLS models. Table 3 presents the regression results. Columns (1) and (2) report the estimated coefficients on pro-environmental attitudes, while columns (3) to (4) report the estimated coefficients on pro-environmental behaviors. As explained above, each model includes a constant, cohort fixed effects, and city dummies, the coefficients of which are not reported for brevity. At the bottom of the table, we report the number of observations and adjusted R-squared. When no control variables are included in columns (1) and (3), we find that education has a significantly positive coefficient estimate. As expected, the estimate of education is still positive and significant at the 1% level when we control for a set of background characteristics in columns (2) and (4), suggesting that more educated individuals are more environmentally friendly.

Inspecting the control variables yields some interesting observations. Rural residents display less pro-environmental behaviors than people in urban areas. This is probably because urban residents have more opportunities to perceive the importance of environmental issues, such as the information from media publicity, and therefore have higher incentives to conduct pro-environmental behaviors.

To solve the potential endogeneity problem, we also perform an instrumental variable analysis. As shown in Table 4, the coefficients on the instrumental variable, *CSL*, in the first-stage regression are positive and highly significant. This is consistent with our expectation that

**Table 4. The effects of education on pro-environmental attitudes and behaviors, 2SLS results.**

|  | Pro-environmental attitudes | Pro-environmental behaviors |
| --- | --- | --- |
|  | (1) | (2) |
| Education | 0.012** | 0.011** |
|  | (0.005) | (0.004) |
| Controls | Yes | Yes |
| Observations | 3,456 | 3,661 |
| **First-stage estimation** |  |  |
| CSL | 2.937*** | 3.141*** |
|  | (0.258) | (0.262) |
| F-statistics | 129.41 | 143.44 |

Note: The table reports the estimates from 2SLS regressions of education on pro-environmental attitudes and behaviors. Columns (1) and (2) report the results for pro-environmental attitudes and for pro-environmental behaviors, respectively. All regressions have the same controls as in column (2) of Table 3, but their coefficients are suppressed for brevity. Robust standard errors are clustered at the cohort-city level and are reported in parentheses.

***, ** and * indicate 1%, 5% and 10% significance levels, respectively.

individuals that are more affected by the CSL have higher educational attainment. The F-test statistics of the first-stage regressions are 129.41 and 143.44 respectively, which exceed the weak instrument critical value established by Staiger and Stock [43]. Taken together, these results indicate that the weak identification is unlikely an issue in our study. More importantly, education remains highly significant in the second-stage regression. The magnitude of the coefficient is not negligible. Examining the economic significance of the education coefficient in column (1), we find that an extra year of schooling is associated with an increase of 0.012 in the pro-environmental attitudes, which is equivalent to 2.1% (= 0.012/0.569) of the sample mean. The coefficient estimate in column (2) implies that an additional year of schooling results in an increase of 0.011 in the pro-environmental behaviors, which is approximately 2% (= 0.011/0.539) of the corresponding sample means. Therefore, the impact of education on pro-environmental attitudes and behaviors is economically sizable on annual basis. Our results show that the IV estimates are larger than the corresponding OLS estimates for regressions of pro-environmental attitudes and behaviors. The downward bias of OLS estimates might be driven by measurement error in survey response. For example, some studies suggest that measurement errors in schooling can lead to downward bias in the OLS estimate of the effect of schooling [29,44].

## 4.2. Heterogeneity analysis

The benchmark results show that educational attainment has strong and positive impacts on the pro-environmental attitudes and behaviors of the whole sample. However, these impacts might differ across different groups of people. In this section, we examine the heterogeneity in the effects of education in the following three aspects. Specifically, we split our sample based on a given individual characteristic (e.g., gender) and then within each subsample, regress pro-environmental attitudes and behaviors on education, controlling for other observable characteristics.

First, we examine whether the education effects on pro-environmental attitudes and behaviors differ by gender. In Table 5, the results in columns (1) and (7) are for male group, while those in columns (2) and (8) are for female group. For pro-environmental attitudes, the estimated coefficient on education is significant for males, but insignificant for females, suggesting that men are more likely to have pro-environmental attitudes through education. This might

**Table 5. Heterogeneity analysis.**

| | Pro-environmental attitudes | | | | | | Pro-environmental behaviors | | | | | |
|---|---|---|---|---|---|---|---|---|---|---|---|---|
| | Male | Female | Urban | Rural | High income | Low income | Male | Female | Urban | Rural | High income | Low income |
| | (1) | (2) | (3) | (4) | (5) | (6) | (7) | (8) | (9) | (10) | (11) | (12) |
| Education | 0.024*** | 0.004 | 0.003 | 0.035*** | 0.011 | 0.023** | 0.002** | 0.015** | 0.006 | 0.023** | 0.013 | 0.012* |
| | (0.009) | (0.005) | (0.007) | (0.011) | (0.012) | (0.011) | (0.001) | (0.006) | (0.006) | (0.009) | (0.011) | (0.007) |
| Controls | Yes | Yes | Yes | Yes | Yes | Yes | Yes | Yes | Yes | Yes | Yes | Yes |
| Observations | 1,646 | 1,810 | 1,807 | 1,649 | 718 | 656 | 1,734 | 1,928 | 1,876 | 1,786 | 753 | 712 |
| **First-stage estimation** | | | | | | | | | | | | |
| CSL | 2.831*** | 3.148*** | 2.658*** | 3.040*** | 2.076*** | 3.620*** | 2.831*** | 3.468*** | 2.828*** | 3.349*** | 2.346*** | 3.812*** |
| | (0.499) | (0.375) | (0.330) | (0.535) | (0.622) | (0.837) | (0.469) | (0.372) | (0.342) | (0.555) | (0.593) | (0.811) |
| F-statistics | 32.12 | 70.50 | 64.72 | 32.19 | 11.12 | 18.67 | 36.55 | 86.92 | 68.25 | 36.34 | 15.62 | 22.08 |

Note: The table reports the results of heterogeneity analysis. The sample is split into two subsamples according to whether the individual is male in columns (1)-(2) and in columns (7)-(8); whether the individual lives in urban areas in columns (3)-(4) and in columns (9)-(10); and whether the individual earns high income in columns (5)-(6) and in columns (11)-(12). All regressions use 2SLS models for estimation and have the same controls as in column (2) of Table 3 except that the variable used as the criteria for splitting the sample is not included. The coefficients on the control variables are suppressed for brevity. Robust standard errors are clustered at the cohort-city level and are reported in parentheses.

***, ** and * indicate 1%, 5% and 10% significance levels, respectively.

be attributed to the gender differences in pro-environmental attitudes that females are found to be more environmentally friendly than males [26]. For pro-environmental behaviors, however, the effects of education attainment do not display gender differences.

Second, given the huge urban-rural divide in China, it is interesting to study whether the education effects are different between these two areas. Columns (3) and (9) present the results for urban residents, while columns (4) and (10) present the results for rural residents. The coefficient on education is positive and significant for rural residents but insignificant for urban residents. There are two possible explanations. First, rural residents exhibit less pro-environmental attitudes and behaviors than those urbanites, as found in our benchmark analysis. Second, rural residents are less exposed to the discussion about environmental issues, e.g. less access to social media. As a result, education plays a bigger role for these individuals.

Finally, we investigate whether the education effects on pro-environmental attitudes and behaviors vary with income levels. To explore this possible heterogeneity, we divide our sample into two distinct income groups, namely the groups of respondents whose income is in the top quintile and bottom quintile of the income distribution. Then we re-run the regressions separately for these two income groups. The results are presented in columns (5)-(6) and (11)-(12) of Table 5. The estimate of schooling years is positive and significant for people in low-income groups, but insignificant for those with high income. Prior studies show that people's demand for public goods such as clean environment will increase with income [45]. Accordingly, high income people might already have a high level of environmental awareness, so that the effect of additional schooling years become negligible. Put it in another word, our finding suggests that income can substitute for the effect of education.

## 4.3. Robustness checks

**4.3.1. Placebo test.** The validity of our instrument relies on the assumption that the CSL influences pro-environmental attitudes and behaviors only through its impact on education rather than through other pathways. Put differently, our instrument should not be systematically correlated with the omitted variables that also affect pro-environmental attitudes and

**Table 6. Robustness checks.**

| | Placebo test | | Alternative pro-environmental attitudes and behaviors | | Sample with shorter birth cohort span | |
|---|---|---|---|---|---|---|
| | Pro-environmental attitudes | Pro-environmental behaviors | Pro-environmental attitudes | Pro-environmental behaviors | Pro-environmental attitudes | Pro-environmental behaviors |
| | (1) | (2) | (3) | (4) | (5) | (6) |
| Education | 0.146 | 0.145 | 0.043** | 0.042** | 0.022** | 0.009** |
| | (1.104) | (0.108) | (0.017) | (0.018) | (0.008) | (0.004) |
| Controls | Yes | Yes | Yes | Yes | Yes | Yes |
| Observations | 3,337 | 3,542 | 3,456 | 3,661 | 2,403 | 2,511 |
| **First-stage estimation** | | | | | | |
| CSL | 1.052 | 1.052 | 2.937*** | 3.141*** | 1.794*** | 1.841*** |
| | (0.693) | (0.693) | (0.258) | (0.262) | (0.283) | (0.273) |
| F-statistics | 2.31 | 2.31 | 129.41 | 143.44 | 40.07 | 45.31 |

Note: The table reports the estimates from three robustness checks. 2SLS models are used for estimation. In columns (1)-(2), two placebo tests are conducted based on the assumption that the CSL had been implemented five years earlier than the actual implementation year of each province. In columns (3)-(4), alternative pro-environmental attitudes and pro-environmental behaviors obtained from factor analysis are used as the dependent variables. In columns (5)-(6), the sample is restricted to individuals born less than 16 year earlier than the first CSL-affected cohort. All regressions have the same controls as in column (2) of Table 3, but their coefficients are suppressed for brevity. Robust standard errors are clustered at the cohort-city level and are reported in parentheses. ***, ** and * indicate 1%, 5% and 10% significance levels, respectively.

behaviors. Otherwise, it may just reflect some province-specific structural changes other than the implementation of the CSL that are correlated with pro-environmental attitudes and behaviors. To check the validity of this assumption, we conduct a placebo test. Specifically, we assume that the CSL had been implemented five years earlier than the actual implementation year of each province and re-run the regressions as Eqs (2) and (3) using the false reform exposure as the instrument. We expect that the false reform exposure has no impact on individual education levels and, more importantly, no impact on pro-environmental attitudes and behaviors.

Columns (1) and (2) of Table 6 show the results of the placebo test. Obviously, the coefficients on the false reform exposure are insignificant in both stages of regressions. This finding substantiates the validity of our instrumental variable and indicates that the benchmark results are reliable.

**4.3.2. Alternative pro-environmental attitudes and behaviors.** We further verify whether our benchmark results are robust to alternative measure of pro-environmental attitudes and behaviors constructed by factor analysis. Factor analysis takes into account the correlations between answers to different attitudes/behaviors questions and the random error in measurement [46]. Specifically, we obtain an index of pro-environmental attitudes by performing a factor analysis on the three attitudes questions and an index of pro-environmental behaviors by performing a factor analysis on the six behaviors questions. The two indexes serve as alternative measures of pro-environmental attitudes and behaviors.

We run 2SLS regressions with the alternative measure of pro-environmental attitudes and behaviors as the dependent variable and present the results in columns (3) and (4) of Table 6. Obviously, in both regressions, the estimates of education are positive and highly significant, suggesting that the positive impacts of education on pro-environmental attitudes and behaviors are robust to alternative measure of pro-environmental attitudes and behaviors.

**4.3.3. Restricted sample.** Given that our sample covers a long span of birth cohorts, we check the robustness of our results by considering a sample with shorter birth cohort span. In particular, we restrict our sample to those born less than 16 years earlier than the first CSL-affected cohort. Columns (5) and (6) of Table 6 report the results in the restricted sample. It turns out that the effects of education remain significantly positive.

## 5. Mechanism analysis

After establishing the casual link between education and pro-environmental attitudes and behaviors, it would be interesting to explore the mechanism that drives this strong relationship. In particular, we focus on one important channel, namely environmental knowledge.

The willingness to alleviate environmental problem is a partial function of risk perception, which is reflected in environmental knowledge [5]. The acquisition of environmental knowledge makes individuals more informed about the situation of local and global environment, negative health effects of environmental problems, and public environmental policies. Indeed, previous literature has established that education facilitates a better understanding of complex environmental messages such as climate change [15,19,20]. In other words, the environmental knowledge may play a major role in forming people's pro-environmental attitudes and behaviors. Hence, we postulate that the environmental knowledge may be a channel underlying the relationship between education and pro-environmental attitudes and behaviors. We test this mechanism in the following.

Environmental knowledge refers to people's knowledge of environmental protection and climate literacy [15]. In the CGSS 2010, the respondents are asked ten true or false questions covering fundamental environmentally-related concepts to evaluate their environmental literacy. These questions are expressed as they would be present in daily life and media publicity. The exact wording of questions is provided in Table 7.

**Table 7. Summary of environmental knowledge.**

| | % of respondents | | |
|---|---|---|---|
| | Correct | Incorrect | Do not know |
| Automobile exhaust does not pose a threat to human health. True or False? | 81.2 | 12.4 | 6.4 |
| Abuse of fertilizers and pesticides damages the environment. True or False? | 83.6 | 9.6 | 6.8 |
| Use of phosphorus-containing washing powder does not cause water pollution. True or False? | 62.1 | 12.9 | 25.0 |
| Fluoride emissions from fluorinated refrigerators damages the ozone layer in the atmosphere. True or False? | 51.7 | 9.7 | 38.6 |
| Acid rain has nothing to do with burning coal. True or False? | 44.3 | 10.8 | 44.8 |
| Species depend on each other, and the disappearance of one species has a chain reaction. True or False? | 52.4 | 5.8 | 41.7 |
| In the domestic air quality reports, air of third-level quality is better than air of first-level quality. True or False? | 26.2 | 11.0 | 62.7 |
| A single species of forest is more susceptible to pests and diseases. True or False? | 44.4 | 9.2 | 46.3 |
| In the domestic water pollution reports, water of V (5) quality is better than water of I (1) quality. True or False? | 16.4 | 7.9 | 75.7 |
| The increase in carbon dioxide in the atmosphere is a contributor of global warming. True or False? | 53.2 | 4.9 | 41.9 |

Note: This table reports the proportion of respondents providing correct, incorrect, and "do not know" answers to each of the ten environmental knowledge questions. Percentages may not sum up to 100 due to rounding.

**Table 8. The effects of education on environmental knowledge.**

|  | Environmental knowledge | |
|---|---|---|
|  | OLS | 2SLS |
|  | (1) | (2) |
| Education | 0.191*** | 0.340*** |
|  | (0.013) | (0.065) |
| Controls | Yes | Yes |
| Observations | 3,601 | 3,601 |
| **First-stage estimation** |  |  |
| CSL |  | 3.157*** |
|  |  | (0.260) |
| F-statistics |  | 147.00 |

Note: The table reports the estimates from regressions of education on environmental knowledge. Column (1) uses OLS model for estimation, while column (2) uses 2SLS model for estimation. All regressions have the same controls as in column (2) of Table 3, but their coefficients are suppressed for brevity. Robust standard errors are clustered at the cohort-city level and are reported in parentheses. ***, ** and * indicate 1%, 5% and 10% significance levels, respectively.

Table 7 summaries the responses to the environmental knowledge questions. In particular, it reports the proportion of respondents providing correct, incorrect and "do not know" answers to each of the ten environmental literacy questions. Overall, we observe a lack of environmental literacy among Chinese people. Most respondents could provide correct answers to the questions of automobile exhaust (81.2%) and abuse of fertilizers (83.6%), while the proportions of correct answers to the questions of acid rain, single species, domestic water pollution reports and air quality reports fail to exceed 50%. The question of domestic water pollution reports turns out to be the most difficult question for respondents, as the percentage of correct answer to that question (16.4%) is the lowest among the ten questions. Then, we construct an environmental knowledge score by awarding one point for each correct answer. Thereby the resulting score, *Environmental knowledge*, lies between 0 and 10.

We relate this environmental knowledge score to educational attainment and report the results in Table 8. The coefficients on education are significantly positive, indicating that the level of education has a positive impact on environmental knowledge. This supports the notion that the education effects pass through environmental knowledge.

Furthermore, we investigate the extent to which this mediating variable contributes to the effects of education on pro-environmental attitudes and behaviors. To achieve this goal, we follow the literature [14,39] and perform a standard mediation analysis in which we assess how the coefficient of education changes when the mediating variable is added as controls to the regressions. Specifically, we compute the percent change in the magnitude for the coefficient of education when a particular mediator is added and interpret it as the proportion of effects of education that are explained by that mediator.

Table 9 reports the results of the mediation analysis. We keep the sample in which all the variables are consistently available to ensure that the changes in the estimated coefficients across regressions are not driven by changes in the sample. Consequently, we end up with 3,401 observations for pro-environmental attitudes and 3,601 observations for pro-environmental behaviors. We report the corresponding benchmark estimates without the mediator in the columns (1) and (3) for the purpose of comparison. Columns (2) and (4) show the results when the variable *Environmental knowledge* is included as an explanatory variable. The

**Table 9. The part of education effects mediated by environmental knowledge.**

| | | Pro-environmental attitudes | | Pro-environmental behaviors | |
| --- | --- | --- | --- | --- | --- |
| | **Base** | **Base+ Environmental knowledge** | **Base** | **Base+ Environmental knowledge** |
| | **2SLS** | **2SLS** | **2SLS** | **2SLS** |
| | **(1)** | **(2)** | **(3)** | **(4)** |
| Part of effects can be explained | | 41.6% | | 63.6% |
| Education | 0.012** | 0.007 | 0.011** | 0.004* |
| | (0.004) | (0.005) | (0.004) | (0.001) |
| Environmental | | 0.013*** | | 0.019*** |
| knowledge | | (0.003) | | (0.003) |
| Controls | Yes | Yes | Yes | Yes |
| Observations | 3,401 | 3,401 | 3,601 | 3,601 |
| **First-stage estimation** | | | | |
| CSL | 2.949*** | 2.448*** | 3.154*** | 2.625*** |
| | (0.255) | (0.257) | (0.260) | (0.261) |
| F-statistics | 133.44 | 90.61 | 146.58 | 101.50 |

Note: The table reports the results of mediation analysis. 2SLS models are used for estimation. As the potential mediator, environmental knowledge is added to the regressions in column (2) and in column (4), respectively. For the purpose of comparison, the results of base model without environmental knowledge are reported in column (1) and in column (3). All regressions also control for the covariates in the basic model as in column (2) of Table 3. The coefficients on the control variables are suppressed for brevity. Robust standard errors are clustered at the cohort-city level and are reported in parentheses. ***, ** and * indicate 1%, 5% and 10% significance levels, respectively.

variable *Environmental knowledge* has significant and positive effects on pro-environmental attitudes and behaviors. The addition of the environmental knowledge score in column (2) causes the coefficient of education to decrease from 0.012 to 0.007, indicating that 41.6% of the effect of education is explained by environmental knowledge. Similarly, we add the environmental knowledge score in column (4), resulting in a decrease in the magnitude of the coefficient of education by 63.6%. Moreover, the estimated coefficients of education in columns (2) and (4) even become insignificant after controlling for the environmental knowledge. These results reveal that the mediator of environmental knowledge accounts for a large proportion of the effect of education. Hence, improved environmental knowledge is the important pathway through which education influences pro-environmental attitudes and behaviors.

## 6. Conclusions

In this paper, we investigate whether educational attainment promotes pro-environmental attitudes and behaviors in China, using microdata from CGSS 2010. We first construct an attitude score and a behavior score using the responses to three questions regarding the willingness of their pro-environmental attitudes and six questions about the frequency of their pro-environmental behaviors, respectively. We establish causality by exploiting the exogenous variation induced by the implementation of the Compulsory Schooling Law (CSL) in China. We find evidence that educational attainment is associated with higher levels of both pro-environmental attitudes and behaviors. The estimates from 2SLS regressions reveal that an additional year of schooling results in an increase of 0.012 and 0.011 in the pro-environmental attitudes and behaviors, which are approximately 2.1% and 2% of the corresponding sample means. Moreover, we show that the positive effects of education on pro-environmental attitudes and behaviors are robust to various robustness checks and are heterogeneous across individuals. Furthermore, we focus on environmental knowledge as the major pathway underlying the

relationship between education and pro-environmental attitudes and behaviors. We find that the acquisition of environmental knowledge is the channel that drives the effect of education on pro-environmental attitudes and behaviors.

Our study offers crucial policy implications as well. Chinese government has been seeking to set the country on a path to more sustainable development, focusing on the effectiveness administrative and regulatory instruments [30]. Our study provides an alternative perspective. Specifically, increasing access to general education could be an essential and effective measure to motivate pro-environmental attitudes and behaviors among a broad population. In particular, the effect of general education in promoting environmental knowledge is crucial for developing countries such as China, where many people have low levels of environmental knowledge.

Our study might have some limitations. First, our study is based on survey responses, which can suffer from measurement error problems. For example, well-educated people may be more likely to over-claim their pro-environmental attitudes and behaviors for keeping a good image. If this is the case, the effect of education on actual pro-environmental attitudes and behaviors might be overstated. In addition, partly due to data availability, we focus on the knowledge acquisition channel. There might exist other (coexisting) channels underlying the relationship between education and pro-environmental attitudes and behaviors. For example, given the strong relationship between educational attainment and labor market outcomes, people with a higher educational attainment might be more optimistic about their future income, and thus have less concern about insufficient material needs when following a pro-environmental lifestyle. Moreover, as in general the exclusion restrictions in IV models are not testable [47], other research methods (e.g., lab and field experiments) can be useful to validate our findings. Furthermore, our study focuses on the effect of schooling education but is salient on the effect of other types of education. We leave these issues to future research.

## Supporting information

**S1 Appendix. Year of implementation of compulsory schooling law across provinces.** (DOCX)

**S2 Appendix. Balancing test.** (DOCX)

## Author Contributions

**Conceptualization:** Geng Niu.

**Data curation:** Qi Wang, Qiaoling Cai.

**Funding acquisition:** Geng Niu.

**Methodology:** Qi Wang, Qiaoling Cai.

**Project administration:** Geng Niu.

**Resources:** Geng Niu.

**Software:** Qi Wang, Xu Gan.

**Supervision:** Geng Niu.

**Writing – original draft:** Xu Gan.

**Writing – review & editing:** Geng Niu.

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
