## [Decision Letter · Decision Letter 0]

12 Jul 2021

PONE-D-21-14017

Green Returns to Education: Does Education Affect Pro-Environmental Attitudes and Behaviors in China?

PLOS ONE

Dear Dr. Gan,

Thank you for submitting your manuscript to PLOS ONE. After careful consideration, we feel that it has merit but does not fully meet PLOS ONE’s publication criteria as it currently stands. Therefore, we invite you to submit a revised version of the manuscript that addresses the points raised during the review process.

Both qualified reviewers have provided constructive comments for a major revision. Please try to address their concerns as much as you can. I also have two additional suggestions. (1) If the data contain city code, you could include city fixed effects instead of province fixed effects as environmental policies and qualities are very city-specific. (2) In table 9, columns 2 and 4, you should report the coefficient of environmental knowledge variable.

We look forward to receiving your revised manuscript.

Kind regards,

Shihe Fu, Ph.D.

Academic Editor

PLOS ONE

Journal Requirements:

Reviewers' comments:

Reviewer's Responses to Questions

**Comments to the Author**

1. Is the manuscript technically sound, and do the data support the conclusions?

Reviewer #1: Partly

Reviewer #2: Yes

2. Has the statistical analysis been performed appropriately and rigorously? 

Reviewer #1: Yes

Reviewer #2: Yes

3. Have the authors made all data underlying the findings in their manuscript fully available?

Reviewer #1: No

Reviewer #2: Yes

4. Is the manuscript presented in an intelligible fashion and written in standard English?

Reviewer #1: Yes

Reviewer #2: Yes

5. Review Comments to the Author

Reviewer #1: This paper aims to estimate the green return to education in China. Using individual data from CGSS 2010, the authors link years of schooling to pro-environmental attitudes and behaviors. To alleviate the endogeneity issue, the authors instrument education attainment with personal exposure to the implementation of the Compulsory Schooling Law in China. Based on a 2SLS model, they find that educational attainment has a moderately positive impact on individual pro-environmental attitudes and behaviors. They further provide evidence that the acquisition of environmental knowledge is one channel that explains the education effect.

Below, I outline my suggestions and comments, which hopefully will improve this paper.

1. The sample selection problem. The authors use the 2010 round of CGSS data originally containing 11,783 respondents residing in 134 cities across 31 of China’s provinces. However, due to the missing values of some variables (what variables? Demographic factors or pro-environmental attitude/behaviors?), the authors drop roughly 70% of the observations without show the geographic distribution of the remaining sample, raising the concern of sample representativeness. Besides, the missing rate of outcome variables may be related to respondents’ education level. For example, surveyed individuals with lower education levels may be less likely to answer questions related to pro-environmental attitudes/behaviors. If this is the case, then the analysis is derived from a selected sample and subject to external validity issues. I would suggest the authors conduct balancing checks based on all observational variables between the regression sample and the sample the author dropped. By doing this, the reader could better understand the representativeness of the regression sample.

2. The potential measurement error in pro-environmental attitude/behaviors. Well-educated people may be more likely to over-claim their pro-environmental attitudes and behaviors for keeping a good image. Then the effect of education may be overstated. The authors may include this point in the discussion/caveat section. Also, in section 4.3.2, the authors use an alternative measure of pro-environmental attitudes/behaviors to conduct a robustness check. However, without a detailed elaboration of how to construct these new measurements, the readers are hardly convinced by this piece of evidence.

3. Specification. The current sample contains a long span of birth cohorts. Apart from the province fixed effects, I would suggest the authors include the cohort fixed effects, which captures the time-varying factors related to the pro-environmental attitudes and behaviors. For perspective, people from different cohorts may have different reporting styles, either over-claim or under-claim. They may have different preferences for cleaner air at the cost of income. They also have different techniques for learning new knowledge. Given a substantial difference among various cohorts, I feel the necessity of including the cohort fixed effects in the OLS as well as the 2SLS regressions.

4. Exclusion restrictions. The authors instrument years of schooling with personal exposure to the implementation of the Compulsory Schooling Law in China (CSL, hereafter). This IV passes the weak instrument test. However, it may not pass the requirement of exclusion restrictions. For example, given a strong relationship between educational attainment and labor outcome, treated people may have a higher expectation of their future income. People who are optimistic about their future without fear of insufficient material needs may pay more attention to environmental protection and pursuit higher quality of life. If this is the case, then CSL may affect people’s pro-environmental attitudes and behaviors through multiple channels, failing to address the endogeneity problem.

5. Heterogeneity. The authors split the sample into top quintile and bottom quintile of the income groups and check the heterogeneity by income. I am wondering whether the observations from the high-income group are mainly coming from urban areas, while those in the low-income group are from rural areas. If that is the case, then the results presented in columns (5)-(6) and (11)-(12) are explained by urban-rural status rather than income level. I would suggest the authors check the heterogeneity by income conditioning on gender or urban-rural status, and these results may provide more insight into the influence of income.

6. Typo: It should be Table 5 rather than Table 6 on page 19 paragraph 2.

Reviewer #2: This paper examines how educational attainment affects pro-environmental attitudes and behaviors in China. It uses cross-section data from CGSS 2010 and exploits the exogenous variation of implementing Compulsory Schooling Law (CSL) in China to identify a causal relationship. The major issue is endogeneity of educational attainment. Its identification strategy is to instrument education attainment by a variable measuring one’s exposure to CSL. The identifying assumption is that the implementation of CSL is exogenously determined and would not directly affect pro-environmental attitudes and behaviors at adulthood. The estimates show that an additional year of schooling results in an increase of 0.084 in the pro-environmental attitudes and 0.219 in the pro-environmental behaviors.

The manuscript is well-polished, but I have a number of comments which may improve this paper.

1. Contribution and the introduction part

There is abundant literature documenting the relationship between education attainment and social attitudes or behaviors related to environmental protection. The marginal contribution of this article is to deal with the endogeneity of education in this context. How can we compare the findings of this paper with previous literature? Are the estimates of education attainment different with studies which fail to deal with endogeneity problem or use data from other countries? It is important to understand how different or unique the findings are when we focus on China. A more comprehensive literature review could be helpful.

Moreover, is there any theory to predict the relationship between education attainment and pro-social behaviors? Can education attainment negatively affect pro-environmental behaviors or they have a non-linear relationship? The authors could refer to a broader scope of literature and motivate the research question more carefully.

2. Birth cohort fixed effects

The age range of respondents in the sample is large. Respondents from different cohorts vary greatly in their social attitudes. By including the age variable may not be enough to control unobserved heterogeneity across birth cohorts. As individuals exposed to CSL are from younger sample, I’m a bit concerned that the major finding that more education attainment promotes pro-environmental attitudes is actually driven by birth cohort variations. I suggest that the authors control cohort fixed effects in the regression, for example, constructing dummies for every 10-year age interval.

3. Interpretation of OLS and IV estimates

According to Table 3 and Table 4, we know that if we ignore the endogeneity problem of education OLS will overestimate the effect on pro-environmental attitudes but underestimate that on pro-environmental behaviors. How can we interpret this? Please carefully discuss how OLS estimates and IV estimates are different and why.

4. Alternative mechanism of the findings

To interpret the major findings, the authors test the environmental knowledge mechanism among others. The mediation analysis shows that more years of education improves environmental knowledge, hence promoting pro-environmental attitudes and behaviors. But an alternative mechanism could be that individuals with more education attainment have higher incomes, so the marginal utility of pro-social behaviors relative to marginal utility of money is larger for them than that for the poor. As a matter of fact, the authors already did a heterogeneity analysis by splitting the high-income and the low-income. From Table 5 we can know that the positive effect of education attainment is mainly driven by the low-income group. This implies that it is environmental knowledge which plays a role, not marginal utility of alleviating environmental problems. The authors could discuss the mechanisms in more detail here.

5. Scale of measures

It is not very clear why and how the authors construct the attitudes questions into a variable ranging from 1 to 15 and the behaviors questions into a variable ranging from 1 to 18. Why not construct two variables of the same range, or just use the original survey questions as dependent variables? Are the findings robust against methods of constructing dependent variables?

In addition, I suggest that maximum and minimum of all variables could also be reported in Table 2.

6. Missing sample

The sample used in this analysis is 3,661 observations, around 30% of the full CGSS 2010 sample. Are the respondents randomly chosen to answer the pro-environmental questions? More explanation how the sample is constructed could be included.

6. PLOS authors have the option to publish the peer review history of their article (what does this mean?). If published, this will include your full peer review and any attached files.

Reviewer #1: No

Reviewer #2: No

---

## [Author Response · Author response to Decision Letter 0]

13 Oct 2021

We have uploaded our response to specific reviewer and editor comments as a separate file labeled "Response to Reviewers". Please find this attached file.

---

## [Decision Letter · Decision Letter 1]

26 Nov 2021

PONE-D-21-14017R1Green Returns to Education: Does Education Affect Pro-Environmental Attitudes and Behaviors in China?PLOS ONE

Dear Dr. Gan,

Thank you for submitting your manuscript to PLOS ONE. After careful consideration, we feel that it has merit but does not fully meet PLOS ONE’s publication criteria as it currently stands. Both the reviewers still raised some minor concerns. Therefore, we invite you to submit a revised version of the manuscript that addresses the points raised during the review process.

We look forward to receiving your revised manuscript.

Kind regards,

Shihe Fu, Ph.D.

Academic Editor

PLOS ONE

Journal Requirements:

Reviewers' comments:

Reviewer's Responses to Questions

**Comments to the Author**

1. If the authors have adequately addressed your comments raised in a previous round of review and you feel that this manuscript is now acceptable for publication, you may indicate that here to bypass the “Comments to the Author” section, enter your conflict of interest statement in the “Confidential to Editor” section, and submit your "Accept" recommendation.

Reviewer #1: (No Response)

Reviewer #2: All comments have been addressed

2. Is the manuscript technically sound, and do the data support the conclusions?

Reviewer #1: Yes

Reviewer #2: Yes

3. Has the statistical analysis been performed appropriately and rigorously? 

Reviewer #1: Yes

Reviewer #2: Yes

4. Have the authors made all data underlying the findings in their manuscript fully available?

Reviewer #1: (No Response)

Reviewer #2: (No Response)

5. Is the manuscript presented in an intelligible fashion and written in standard English?

Reviewer #1: Yes

Reviewer #2: Yes

6. Review Comments to the Author

Reviewer #1: (No Response)

Reviewer #2: The current manuscript addresses my comments well by providing a more comprehensive literature review, including richer background information and conducting more rigorous regression analysis. However, there are several comments that the authors can take into further account before making it to a publishable version.

1. page 8, line 6: “…but also because China’s unique culture and institution help validate...”

page 11, line 18: “In addition, China has its own unique culture and ideology, …”

It is not very clear which part of China’s unique cultural or institutional features would help validate the relationship between education and environmental attitudes. The authors should avoid too general arguments without further elaboration in a scientific research paper. Instead, the authors could briefly discuss China’s environmental problems as a consequence of fast industrial growth as part of motivation. That is a unique feature of the Chinese context.

2. page 11, line 5: “…in particular, China, schooling education is examination-oriented and emphasizes competition… are therefore less caring for the environment”

page 23, line 8: “people that are obsessed with personal achievement might care less for the environment.”

There is no solid evidence to support the negative association between education and pro-environmental attitudes. The authors should elaborate very carefully how willingness to compete conflicts with pro-environmental attitudes, and why this is unique in China. As a matter of fact, I think there is no need to raise this point. The downward bias of OLS estimates might be driven by measurement error. I would prefer not to mention this channel.

3. Page 14: why are only those born in February, September, November or December asked about environmental attitudes and behaviors in CGSS? It is difficult to understand.

4. typos

page 7, line 6: “While there are large number of papers…” should be “there are a large number of …”

page 10, line 13: “… skills accumulated through education make individuals be familiar with…”, delete “be”

page 10, line 15: “… people with higher level of formal education…”, should be “with a higher level”

7. PLOS authors have the option to publish the peer review history of their article (what does this mean?). If published, this will include your full peer review and any attached files.

Reviewer #1: No

Reviewer #2: No

---

## [Author Response · Author response to Decision Letter 1]

3 Dec 2021

We have revised the manuscript to carefully address the editor's and reviewers’ insightful comments. Please see our revised manuscript and responses to editor's and reviewers’ comments for details.

---

## [Editor Report · Decision Letter 2]

19 Jan 2022

Green Returns to Education: Does Education Affect Pro-Environmental Attitudes and Behaviors in China?

PONE-D-21-14017R2

Dear Dr. Gan,

I see you have addressed the referees' comments well in this second-round revision and there is no need to send out for review again. We’re pleased to inform you that your manuscript has been judged scientifically suitable for publication and will be formally accepted for publication once it meets all outstanding technical requirements.

Kind regards,

Shihe Fu, Ph.D.

Academic Editor

PLOS ONE
---

## [Editor Report · Acceptance letter]

26 Jan 2022

PONE-D-21-14017R2 

Green Returns to Education: Does Education Affect Pro-Environmental Attitudes and Behaviors in China? 

Dear Dr. Gan:

I'm pleased to inform you that your manuscript has been deemed suitable for publication in PLOS ONE. Congratulations! Your manuscript is now with our production department. 

Kind regards, 

on behalf of

Dr. Shihe Fu 

Academic Editor

PLOS ONE